# Surface tension and hygroscopicity analysis of aerosols containing organosulfate surfactants

Vahid Shahabadi <sup>1</sup>, Cassandra Lefort <sup>2</sup>, Hoi Tang Law <sup>3</sup>, Man Nin Chan <sup>3</sup>, and Thomas C. Preston <sup>1,2</sup>

**Correspondence:** Thomas C. Preston (thomas.preston@mcgill.ca)

#### Abstract.

Organosulfate (OS) surfactants can influence cloud condensation nuclei (CCN) activation and hygroscopic growth by reducing the surface tension of aerosol particles. We investigate the surface tension and hygroscopicity of aerosols containing shortand long-chain OSs in supersaturated aqueous droplets using an electrodeformation method coupled with Raman spectroscopy. For droplets containing short-chain OSs, the surface tension decreases as relative humidity (RH) decreases, even under dry and highly viscous conditions. Sodium ethyl sulfate (SES) lowered surface tension to approximately 30 mN m<sup>-1</sup>, a value lower than that of sodium dodecyl sulfate (SDS) at its critical micelle concentration. We also studied ternary systems containing OSs with citric acid (CA) or sodium chloride (NaCl). Even small amounts of SDS, with a molar ratio of 10<sup>-3</sup> relative to CA, reduce surface tension by up to 40% at low RH compared to CA alone. Despite strong surface tension reduction, ternary OS–CA–water systems show hygroscopicity nearly identical to binary CA–water systems, suggesting that surface tension does not influence water uptake under subsaturated conditions. Ternary systems containing NaCl and OS undergo efflorescence at 47% RH, but the crystallized NaCl becomes partially engulfed. If the RH is subsequently increased, the particle takes up water. At the deliquescence point (72% RH), the particle becomes homogeneous again. These findings improve our understanding of particle growth and cloud drop formation processes, which influence cloud properties like albedo and lifetime.

## 15 1 Introduction

Atmospheric aerosols exert cooling effects on Earth's climate, both directly by scattering incoming solar radiation, and indirectly by serving as cloud condensation nuclei (CCN), which influences cloud albedo and precipitation patterns (Haywood and Boucher, 2000; Yu et al., 2006; Heald et al., 2014; Rap et al., 2013; Scott et al., 2014). Predicting the climate effects of atmospheric aerosols remains a significant challenge in climate models. Indeed, the aerosol indirect effect, particularly the cloud-albedo effect, is considered the largest source of uncertainty in radiative forcing estimates (Stocker et al., 2013; Intergovernmental Panel on Climate Change, 2023). This uncertainty stems largely from difficulties in predicting the fraction of aerosols that activate as CCN (Carslaw et al., 2013; Lee et al., 2016; Vogel et al., 2022). Köhler theory is the standard framework for predicting CCN activation (Köhler, 1936). According to Köhler theory, an aerosol particle activates into a cloud

<sup>&</sup>lt;sup>1</sup>Department of Atmospheric and Oceanic Sciences, McGill University, Montreal, Quebec, Canada

<sup>&</sup>lt;sup>2</sup>Department of Chemistry, McGill University, Montreal, Quebec, Canada

<sup>&</sup>lt;sup>3</sup>Department of Earth and Environmental Science, Faculty of Science, The Chinese University of Hong Kong, Hong Kong, Hong Kong SAR, China

droplet when the ambient water vapor supersaturation exceeds a critical value  $(S_c)$ . This  $S_c$  depends on the particle's size, solute content, and the droplet's surface tension.

While many climate models assume activating aerosol droplets have the surface tension of pure water, this assumption can be invalid for organic-rich aerosols, phase-separated particles (Ovadnevaite et al., 2017), or those containing surfactants (Nozière et al., 2014). Surfactants partition to the aerosol surface, forming monolayers or multilayers, thereby reducing the surface tension below that of pure water (Malila and Prisle, 2018). Furthermore, the high surface-to-volume ratio of aerosols can cause aerosol surface tension to deviate from bulk solution values due to enhanced surface partitioning, leading to higher surface concentrations and depleted bulk concentrations compared to macroscopic solutions (Bzdek et al., 2020). Therefore, assuming pure water surface tension for surfactant-containing aerosols leads to an overestimation of the critical supersaturation required for activation (Li et al., 1998; Sorjamaa et al., 2004) and potentially an underestimation of the CCN population (Ovadnevaite et al., 2017). Accounting for surface tension reduction by surfactants could decrease estimates of the aerosol indirect forcing by up to 1.1 W m<sup>-2</sup>, a magnitude comparable to the total forcing itself (Facchini et al., 1999; Prisle et al., 2012).

Organosulfates (R-OSO<sub>3</sub><sup>-</sup>, OSs) are among the primary and most abundant anionic surfactants in atmospheric aerosols and have been detected worldwide, from North and South America to Asia (Brüggemann et al., 2020; Hettiyadura et al., 2017; Wang et al., 2021). They can be formed through secondary pathways such as radical-initiated formation (Nozière et al., 2010), reactions between derivatives of biogenic volatile organic compounds (BVOCs, e.g., isoprene) and sulfate aerosol (Darer et al., 2011; Iinuma et al., 2009), or sulfate esterification (Minerath et al., 2008). While inorganic sulfates have traditionally been considered the dominant form of atmospheric sulfur, field studies reveal the widespread presence of organosulfur compounds (including OSs, sulfones, sulfonates, and sulfoxides) in aerosols across diverse environments (Tolocka and Turpin, 2012; Shakya and Peltier, 2013). OSs are significant atmospheric surfactants, sometimes constituting up to 30% of the organic aerosol mass (Brüggemann et al., 2020; Fan et al., 2022). For example, sodium dodecyl sulfate (SDS), often used as a model surfactant in aerosol studies, has been extensively studied (Prisle et al., 2010; Gantt and Meskhidze, 2013). OSs derived from BVOCs like isoprene, limonene, and α-pinene, as well as their oxidation products (e.g., methacrolein, methyl vinyl ketone), are also known to be surface-active (Nozière et al., 2010; Hansen et al., 2015; Peng et al., 2021; Xiong et al., 2023). Pure OS aerosols exhibit growth factors around 1.8 at high relative humidity (RH), with hygroscopicity parameter ( $\kappa$ ) values ranging from approximately 0.2 (long-chain) to 0.46 (short-chain) (Peng et al., 2021). Bulk-surface partitioning and bulk depletion are particularly significant in small aerosol droplets containing strong surfactants due to their high surface-to-volume ratio, leading to deviations between aerosol and bulk solution surface tension (Bain et al., 2023b; Schmedding and Zuend, 2023). Bulksurface partitioning becomes dominant at high surface-to-volume ratios. Furthermore, droplet size can influence the effective critical micelle concentration (CMC) of surfactants, thereby affecting the system's surface tension (Jacobs et al., 2024).

50

55

Understanding OS behavior in concentrated, metastable aerosols is important because such conditions are relevant in the atmosphere, particularly at RH below the deliquescence point (typically < 80%). Atmospheric models like the GEOS-Chem chemical transport model (using ISORROPIA II) and the Community Multiscale Air Quality (CMAQ) model account for aerosol thermodynamics in both stable and metastable regimes (Kelly et al., 2010; Miller et al., 2024). Studying the surface tension of metastable aerosols also aids in assessing and refining semi-empirical surface tension models (Kleinheins et al.,

2023, 2024). Hygroscopicity studies show that many OS aerosols (e.g., derived from limonene, other aliphatic/aromatic OSs) often do not exhibit sharp deliquescence or efflorescence transitions (Brüggemann et al., 2020). At low RH, they can exist as highly concentrated aqueous droplets, potentially phase-separated, and may become highly viscous or even glassy, which can impact mass transfer and particle growth (Kwamena et al., 2010).

Currently, most surface tension measurements for OSs, whether in bulk solutions or aerosols, are performed under subsaturated conditions (water activity  $a_w > 0.8$ ), with limited data available for highly concentrated, supersaturated states (Gen et al., 2023; Bain et al., 2023b; Bain, 2024). Bulk-aerosol deviations are less pronounced for weak surfactants at low concentrations (e.g., < 20 mM), where bulk and aerosol measurements often agree (Bain et al., 2023b). However, even weak surfactants like sodium ethyl sulfate (SES) can exhibit significant surface activity in concentrated aerosols (e.g., > 1 M) (Bain et al., 2023a). For SES, while surface tension is near that of pure water below 40 mM, a 4.4–7.8% reduction relative to pure water was observed above 1 M, suggesting different surface behavior in concentrated aerosols compared to dilute bulk solutions. Among the various techniques for measuring aerosol surface tension, Atomic Force Microscopy (AFM) (Lee and Tivanski, 2021), Holographic Optical Tweezers (HOT) (Bzdek et al., 2016), and quasielastic light scattering (OELS) (Gen et al., 2023) are precise methods used for aerosol surface tension measurements. These techniques offer high spatial resolution and the ability to probe single particles or small droplets, making them particularly useful for studying microdroplets relevant to atmospheric processes. However, challenges remain, particularly in measuring supersaturated aerosols at low RH where high viscosity and potential phase transitions can interfere with measurements or particle stability. This is particularly relevant for OS-containing aerosols, whose properties change significantly upon transitioning to highly viscous or glassy states at low RH. Despite its importance, experimental data on highly viscous, concentrated aerosols containing organosulfates remain scarce due to the aforementioned measurement challenges.

In this study, we measure physicochemical properties (density, refractive index, water activity) of aqueous solutions containing short-chain (sodium methyl sulfate, SMS; sodium ethyl sulfate, SES) and long-chain (sodium octyl sulfate, SOS; sodium decyl sulfate, SDeS; sodium dodecyl sulfate, SDS) organosulfates. Using an electrodeformation technique coupled with optical trapping and Raman spectroscopy (Shahabadi et al., 2024), we measure the surface tension of single levitated droplets containing binary (OS + water) and ternary (OS + solute + water) mixtures, focusing on supersaturated conditions (low RH). For comparison at higher water activities (lower concentrations), we use the pendant drop method for bulk solution measurements (Berry et al., 2015). We also investigate the hygroscopic growth of ternary aerosol particles (diameter 5-10 µm) containing citric acid (CA) and trace amounts of OSs. Finally, we examine the efflorescence and deliquescence behavior of ternary systems containing sodium chloride (NaCl) and trace OSs.

## 2 Methods

#### 2.1 Materials

Measurements were conducted on aqueous solutions of five sodium alkyl sulfates: sodium methyl sulfate (SMS; Sigma,  $\geq$  92%), sodium ethyl sulfate (SES; Sigma,  $\geq$  98%), sodium octyl sulfate (SOS; Sigma,  $\geq$  95%), sodium decyl sulfate (SDeS;

Sigma,  $\geq$  99%), and sodium dodecyl sulfate (SDS; Sigma,  $\geq$  99%). For investigations involving ternary systems, sodium chloride (NaCl; Sigma,  $\geq$  99%) and citric acid (Fisher Chemicals, 100%) were used. All solutions were prepared using deionized water.

# 5 2.2 Physicochemical property measurements

Aqueous solutions of each solute were prepared at various concentrations spanning the range relevant for subsequent surface tension measurements. The density  $(\rho)$  of each bulk solution was determined gravimetrically by weighing 5 mL aliquots dispensed with a micropipette (Sartorius AG, Germany) on a digital balance (Sartorius AG, Germany) accurate to  $10^{-4}$  mg. The refractive index (n) was measured at the sodium D-line (589 nm) using an Abbe refractometer (Fisher Scientific Co., Japan) with an accuracy of  $\pm 0.0002$ . The water activity  $(a_{\rm w})$  of each solution was determined using a water-activity meter (AquaLAB 4TE, METER Group, Inc.), which provides an accuracy of  $\pm 0.003$ . Approximately 5 mL of each solution was placed in a polystyrene petri dish, sealed within the instrument's chamber, and allowed to equilibrate before measurement.

Density, refractive index, and water activity measurements were performed at 295 K and data were fitted using standard empirical expressions (Lienhard et al., 2012). For density,

105 
$$\rho = \rho_0 + aw + bw^2$$
, (1)

where  $\rho_0$  is the density of pure water, w represents the mass fraction of the solute, and a and b are fitting parameters. For refractive index,

$$n = n_0 + cn_1 + c^2 n_2, (2)$$

where  $n_0$  is the refractive index of pure water, c is the molar concentration, and  $n_1$  and  $n_2$  are fitting parameters. For water activity,

$$a_{\rm w} = \frac{1 - w}{1 + qw + rw^2},\tag{3}$$

where q and r are dimensionless fitting parameters. The best-fit parameters for aqueous solutions of five sodium alkyl sulfates are listed in Table 1.

Surface tension ( $\sigma$ ) was measured using two complementary techniques depending on the solution concentration and corresponding water activity: the pendant drop method for lower concentrations (higher  $a_{\rm w}$ ) and electrodeformation of optically trapped droplets for higher concentrations (lower  $a_{\rm w}$ ). The details of these methods are given in Sections 2.2.1 and 2.2.2, respectively.

The surface tension measurements of binary solutions were modeled as a function of the amount of solute using (i) the Szyszkowski–Langmuir model (von Szyszkowski, 1908; Langmuir, 1917) and (ii) the Eberhart model (Eberhart, 1966; Kleinheins et al., 2023). The Szyszkowski–Langmuir model relates the surface tension of the system ( $\sigma$ ) to that of pure water ( $\sigma$ <sub>w</sub>), temperature (T), and molar concentration (c) of the solute through:

$$\sigma = \sigma_{\rm w} - RT\Gamma_{\infty} \ln(1 + c/\beta),\tag{4}$$

| Solute | Molar Mass (g mol <sup>-1</sup> ) | $\rho$ Fit (Eq. 1) |                                     | n Fit (Eq. 2)                                     |                                     | $a_{\mathrm{w}}$ Fit (Eq. 3) |           |
|--------|-----------------------------------|--------------------|-------------------------------------|---------------------------------------------------|-------------------------------------|------------------------------|-----------|
|        |                                   | $a (g mL^{-1})$    | $b  (\mathrm{g}  \mathrm{mL}^{-1})$ | $n_1  (\operatorname{L} \operatorname{mol}^{-1})$ | $n_2 (L^2 \operatorname{mol}^{-2})$ | q                            | r         |
| SMS    | 134.08                            | 0.4983             | 0.2665                              | 0.01104                                           | -0.0005972                          | -0.9234                      | 0.6407    |
| SES    | 148.10                            | 0.3760             | 0.3909                              | 0.01209                                           | -0.0006125                          | -0.9689                      | 0.4481    |
| SOS    | 232.28                            | 0.2612             | -0.1565                             | 0.02605                                           | -0.002604                           | -0.9592                      | 0.002489  |
| SDeS   | 260.33                            | 0.05800            | 0.2762                              | 0.03679                                           | -0.008359                           | -0.9767                      | 0.0005819 |
| SDS    | 288.38                            | 0.1276             | -0.04706                            | 0.02738                                           | 0.007907                            | -0.9756                      | 0.01754   |

**Table 1.** Parameters for density  $(\rho)$ , refractive index (n) at  $\lambda = 589$  nm, and water activity  $(a_{\rm w})$  models according to Equations 1, 2, and 3. Note that  $\rho$  and  $a_{\rm w}$  models use mass fraction (w), while the n model uses molar concentration (c). Units for  $\rho$  are in g mL<sup>-1</sup>, n is dimensionless,  $a_{\rm w}$  is dimensionless, c are in mol L<sup>-1</sup>. All measurements used for fits were performed at 295 K. For Eq. 1,  $\rho_0$  was set to 1 g mL<sup>-1</sup>, and for Eq. 2,  $n_0$  was set to 1.333.

where  $\Gamma_{\infty}$  is the maximum surface excess concentration and  $\beta$  is related to adsorption equilibrium constant. Both of these terms are fitting parameters. R is the universal gas constant. Here,  $\sigma_{\rm w}$  = 72.5 mN m<sup>-1</sup>, which is the surface tension of water at T = 295 K.

The Eberhart model relates surface tension to the bulk mole fraction of the solute  $(x_i)$ :

$$\sigma = \frac{\sigma_{\mathbf{w}}(1 - x_i) + Sx_i\sigma_i}{1 - x_i + Sx_i},\tag{5}$$

where S (surface enrichment factor) and  $\sigma_i$  (surface tension of pure solute component, considered to be hypothetical) are fitting parameters.

# 130 2.2.1 Pendant Drop Tensiometry

125

The pendant drop method (Berry et al., 2015), relying on image analysis of drop shape, was employed for surface tension measurements of bulk solutions. The experimental setup consisted of an LED illumination source (470 nm, ThorLabs, US) and a digital camera (Kiralux 5.0 MP, ThorLabs, US) focused on a pendant drop. Drops were formed at the tip of a syringe needle (EXEL International, CA), with the diameter of 1.65 mm, containing the solution of interest. Once a stable drop was formed, images were captured. The typical size range of the pendant drop was approximately 2–3 mm at the lower portion of the drop.

Surface tension was calculated from the drop profile using the open-source software ImageJ (Collins, 2007; imagej.net) equipped with a "pendant drop" analysis plugin. The software iteratively fits the Young-Laplace equation to the digitized drop profile to determine the Bond number (Bo), a dimensionless shape parameter. The surface tension is then calculated via:

$$\sigma = \frac{\Delta \rho g R_0^2}{\text{Bo}},\tag{6}$$

where  $\Delta \rho$  is the density difference between the solution and the surrounding air, g is the acceleration due to gravity (9.8 m s<sup>-2</sup>), and  $R_0$  is the radius of curvature at the drop apex. To ensure accurate Bo determination, drop volumes were maintained sufficiently large such that the Worthington number, Wo =  $V_d/V_{max}$  (where  $V_d$  is the drop volume and  $V_{max}$  is the maximum

stable volume), was close to unity. Measurements were performed within a few seconds of drop stabilization to minimize potential dynamic surface tension effects or artifacts due to evaporation.

# 145 2.2.2 Electrodeformation of trapped droplets



For single-particle measurements, we used a recently developed non-contact electrodeformation technique, to precisely measure the surface tension of supersaturated and viscous aerosols (Shahabadi et al., 2024). This method involves Raman spectroscopy of an optically trapped droplet undergoing deformation induced by an external electric field. The experimental setup comprises a dual-beam optical trap that levitates a micron-sized droplet (5-10  $\mu$ m in diameter) between two electrodes capable of generating a strong electric field. Raman scattering from the droplet is recorded by a spectrometer, and the Mie Resonance Fitting (MRFIT) algorithm (Preston and Reid, 2015) is used to determine droplet size and refractive index. The electric field induces prolate deformation, causing a characteristic splitting of MDRs. Electrocapillary theory (Lac and Homsy, 2007; Yorulmaz et al., 2009) is then applied to quantify the droplet's surface tension ( $\sigma$ ).

As illustrated in Figure 1(a), a single aqueous solution droplet (e.g. a water droplet with surfactants) was levitated using a counter propagating dual beam optical trap in humidified air whose RH is controlled via relative mixing of dry and wet air flows within the cell using two mass flow controllers (MKS Instruments, US). During measurements, the cell RH can be stably maintained below 90%. Above this level, achieving stable humidity requires airflow rates that dislodge the droplet from the optical trap, making such conditions impractical with this cell design.

To measure surface tension, a pair of needle-shaped copper electrodes (AMTOVL), separated by approximately 150  $\mu$ m, was positioned around the trapped droplet as shown in Figure 1(b). The electrodes were pre-sharpened and ground. Applying a voltage across these electrodes generated a controllable electric field (E), causing the droplet to deform from a sphere into a prolate spheroid due to the difference in dielectric properties between the droplet and the surrounding air.

The extent of deformation was quantified by analyzing the splitting of morphology-dependent resonances (MDRs) in the droplet's Raman spectrum. The surface tension was then calculated using (Lac and Homsy, 2007):

$$\sigma = \frac{9}{16} \frac{\varepsilon_m \varepsilon_0 R_o E^2}{D},\tag{7}$$

where  $\varepsilon_m$  is the relative permittivity of the surrounding medium (air,  $\approx$  1),  $\varepsilon_0$  is the permittivity of vacuum (8.854  $\times$  10<sup>-12</sup> F m<sup>-1</sup>),  $R_o$  is the radius of a sphere with equivalent volume to the deformed droplet (determined from MDR analysis of the unperturbed droplet), E is the magnitude of the applied electric field, and D is the deformation parameter. D is defined as  $D = (R_p - R_e)/(R_p + R_e)$ , where  $R_p$  and  $R_e$  are the semi-major and semi-minor axes of the prolate spheroid, respectively.

The deformation parameter D was obtained from the observed MDR splitting using the relation derived from perturbation theory (Shahabadi et al., 2024):

$$\frac{1}{\lambda_{lm}} = \frac{1}{\lambda_l^{\text{Mie}}} \left[ 1 - \frac{D}{3} \left( 1 - \frac{3m^2}{l(l+1)} \right) \right]. \tag{8}$$

Figure 1. Schematic of the electrodeformation method. (a) Two counter-propagating green lasers (532 nm) simultaneously trap the particle and excite morphology-dependent resonances (MDRs) observed in the Raman scattering. (b) The particle is trapped between a pair of electrodes that generate a strong electric field, E, (up to  $\approx 2.7 \times 10^6 \text{ V m}^{-1}$ ) on the droplet (white arrows indicate field direction). Inset shows a levitated droplet. (c) MDRs appear as sharp peaks superimposed on the spontaneous Raman spectrum (e.g., water O-H stretching band). Without an electric field (left panel, E=0), resonances are degenerate. The electric field induces prolate deformation (right panel, E>0), splitting the MDRs into non-degenerate components.

Here,  $\lambda_l^{\text{Mie}}$  is the wavelength of a specific Mie resonance mode (angular momentum index l) for the non-deformed spherical droplet (measured or calculated at E=0), and  $\lambda_{lm}$  is the wavelength of a specific azimuthal component ( $m \neq 0$ ) within the split multiplet observed under the applied electric field (Figure 1(c)).



The optical trapping setup utilized a  $\lambda=532$  nm continuous-wave laser (Verdi G5, Coherent, Inc.). The beam was spatially filtered, linearly polarized at  $45^{\circ}$ , and split into two orthogonally polarized beams of equal power using a calcite beam displacer (Thorlabs, Inc.). These beams were directed into independent trapping arms via a knife-edge prism mirror (gold-coated, Thorlabs) and expanded before being focused to a common point by two long working distance objective lenses ( $50\times$ , NA = 0.5, WD = 20.5 mm, Plan Apo NIR B, Mitutoyo Corporation, Kawasaki, Japan), creating the optical trap. Raman scattering from the trapped, deformed droplet was collected and analyzed using a spectrometer (IsoPlane SCT-320, Teledyne Princeton Instruments, US).

Figure 2. (a) Density ( $\rho$ ), (b) refractive index (n) at  $\lambda$  = 589 nm, and (c) water activity ( $a_{\rm w}$ ) of binary aqueous organosulfate solutions across a range of molar concentrations (M). Organosulfates shown are sodium methyl sulfate (SMS), sodium ethyl sulfate (SES), sodium octyl sulfate (SOS), sodium decyl sulfate (SDeS), and sodium dodecyl sulfate (SDS). Solid lines represent fits to the experimental data using the parameterizations given in Equations 1-3 with the best-fit parameters listed in Table 1.

## 3 Results and Discussion



# 3.1 Physicochemical properties

The experimental procedure begins with preparing bulk solutions of various OSs: SMS, SES, SOS, SDeS, and SDS, with hydrocarbon chain lengths of 1, 2, 8, 10, and 12 carbons, respectively. Density measurements (Figure 2(a)) reveal that short-chain OSs exhibit higher densities at increased concentrations. Long-chain OSs presented measurement challenges at concentrations above 2 M due to low solubility and micelle formation. Refractive index (RI) measurements (Figure 2(b)) indicate that RI generally increases with hydrocarbon chain length. Water activity  $(a_{\rm w})$  sharply decreases with increasing concentration for short-chain OSs (Figure 2(c)), whereas long-chain OSs maintain high and relatively stable  $a_{\rm w}$  (typically > 0.95), likely due to weaker interactions with water molecules. Fitting curves are generated from models detailed in the Methods section.

For binary aerosols containing long-chain OSs, we can only perform measurements at relatively high  $a_{\rm w}$  and are thus restricted in the range of surface tension measurements. Critical micelle concentration (CMC) values for SOS, SDeS, and SDS at room temperature are approximately 130 mM (Rassing et al., 1973; Yatcilla et al., 1996), 33 mM (Robb, 1969; Király and Dekány, 2001), and 5–8 mM (Flockhart, 1957; Moroi et al., 1974), respectively. Thus, we first compared surface tension measurements for binary short-chain OS solutions (SMS and SES), obtained via bulk (pendant drop) and single-particle (electrodeformation) methods, with data from Bain et al. (2023a).

Figure 3. Surface tension ( $\sigma$ ) as a function of relative humidity (RH) for aqueous droplets containing (a) sodium methyl sulfate (SMS) and (b) sodium ethyl sulfate (SES). Symbols represent measurements from this study: bulk solutions (pendant drop, squares) and single aerosols (electrodeformation, triangles). Literature data from Bain et al., 2023a (circles) are shown for comparison. Shaded regions indicate measurement uncertainty. Data are fitted using Langmuir (solid line) and Eberhart (dashed line) models (see Methods). The top x-axis indicates the corresponding approximate bulk concentration (M) of the surfactant. Green diamonds indicate the refractive index, n, of the trapped droplet at  $\lambda = 589$  nm.

#### 3.2 Surface tension



Figure 3(a) shows our bulk SMS measurements aligning closely with literature data at high  $a_{\rm w}$  (low concentration), while electrodeformation results are consistent and overlap with extrapolated literature data (Bain et al., 2023a). Surface tension decreases with decreasing RH, plateauing around 55 mN m<sup>-1</sup> at RH  $\approx$  40%, similar to moderately surface-active organic acids. Thus, SMS can be classified as a weak surfactant (Kleinheins et al., 2025).

In contrast, SES surface tension measurements (Figure 3(b)) agree reasonably well with literature data at lower concentrations but show no plateau, continuing to steeply decrease, even at low RH. Our data, extending to significantly lower RH than prior studies, suggests the surface composition continues to change, even at very high SES concentrations (> 7 M). The electrodeformation technique successfully extends measurable surface tension down to approximately 30 mN m<sup>-1</sup> at RH = 33%. Surface tension data fits well to Langmuir and Eberhart semi-empirical models (Methods).

Although the electrodeformation experiments conducted here were not designed for quantitative viscosity measurements (each stepwise increase in voltage followed by a Raman measurement requires approximately 1 s, restricting observations to extremely viscous droplets), in our experiments on the aqueous SES system, we observed that when the applied voltage was increased from 0 V to 400 V at RH = 28%, the droplet took roughly 15 s to reach its equilibrium shape. For high-viscosity droplets, the characteristic relaxation time can be approximated as  $\tau \sim \eta R_0/\sigma$ , where  $\eta$  is the dynamic viscosity Power and Reid (2014). Applying this relation to the observed 15 s timescale yields an estimated  $\eta \sim 10^6$  Pa s for the SES droplet under these conditions. Therefore, under dry conditions and warm temperatures (RH = 28% and T = 295 K), SES

**Figure 4.** Surface tension ( $\sigma$ ) as a function of relative humidity (RH) for aqueous droplets containing (a) sodium chloride (NaCl), NaCl:SMS (10:1 molar ratio), and NaCl:SES (10:1 molar ratio), and (b) citric acid (CA), CA:SMS (10:1 molar ratio), and CA:SES (10:1 molar ratio). Measurements use bulk (pendant drop) and single-particle electrodeformation methods. Shaded regions indicate measurement uncertainty. The top x-axis indicates the approximate concentration (M) of the major solute (NaCl or CA) in a hypothetical binary aqueous solution without OS at the corresponding RH. The vertical dashed line marks the RH corresponding to the bulk solubility limit of the major solute. Solid curves are added to guide the eye.

aerosols exhibit a highly viscous state ( $\eta \gtrsim 10^6$  Pa s) where the RI of the trapped droplet can reach 1.55. As can be seen in Figure 3 (b), our measurements reveal a sharp increase in the RI at RH below 35%, which likely coincides with a pronounced rise in droplet viscosity. This behavior is also consistent with previous observations showing that OS aerosols do not exhibit efflorescence/deliquescence transitions (Estillore et al., 2016). At approximately 20% RH, the RI increases past 1.6. Below this RH, intensified light scattering and a compromised quality factor prevent us from observing MDRs and RI measurements are no longer possible.

The impact of inorganic and organic solutes on surface tension was studied by adding either NaCl or CA to droplets with OS surfactants. Measurements maintained a 10:1 molar ratio of the main solute (NaCl or CA) to OS (SMS or SES). Results (Figure 4) demonstrate that surfactants significantly reduce surface tension even at low molar fractions, with their effectiveness highly dependent on hydrocarbon chain length.

For solutions containing NaCl (Figure 4(a)), SMS moderately reduces surface tension (by approximately 10% compared to aqueous NaCl) at low RH. SES provides a more substantial reduction (around 24%) at RH = 60%, indicating greater surface activity. For CA-containing droplets (Figure 4(b)), SES again has a greater impact than SMS in terms of surface tension reduction.


Due to challenges associated with low CMC and high  $a_{\rm w}$ , binary surface tension measurements of long-chain OSs were impractical at low RH. Additionally, NaCl induced instability (i.e., precipitation, phase separation) in aqueous systems with long-chain OSs below RH  $\approx$  80%. Therefore, our analysis of long-chain OSs focused on aqueous systems containing CA with either SOS or SDS.

**Figure 5.** Surface tension (*σ*) of citric acid (CA) droplets with and without long-chain surfactants as a function of relative humidity (RH). Data shown for: Pure CA (squares), CA:SOS (100:1 molar ratio, up triangles), and CA:SDS (1000:1 molar ratio, down triangles). The top x-axis indicates the approximate concentration (M) of CA in a hypothetical binary aqueous solution without OS at the corresponding RH. The vertical dashed line marks the RH corresponding to the bulk solubility limit of CA. Data are presented for both bulk (pendant drop) and single-particle electrodeformation, with shaded regions representing uncertainty.

Figure 5 presents surface tension results for aqueous CA droplets mixed with SOS (100:1) or SDS (1000:1). As RH falls below 80–90%, the surface tension measurements converge toward approximately 40 mN m $^{-1}$  (CA/SOS) and 35 mN m $^{-1}$  (CA/SDS), respectively. These values are close to the reported surface tensions of binary SOS (Yatcilla et al., 1996) and SDS (Mysels, 1986; Santos et al., 2003) solutions at their respective bulk CMCs. Both surfactants significantly suppressed droplet surface tension compared to pure CA. Notably, at RH = 30%, SOS reduced surface tension by  $\approx$  36%, and SDS by  $\approx$  44%. SDS, with a longer hydrocarbon chain, exhibited superior surfactant capability even at a tenfold lower molar ratio, approaching typical surface tensions near the bulk CMC conditions reported in the literature. The marked surface tension reduction suggests surface saturation may occur, potentially enhanced by CA–surfactant synergistic interactions, such as hydrogen bonding between CA's hydroxyl and carboxyl groups and the surfactant sulfate headgroups, partial protonation of CA reducing electrostatic repulsion and allowing tighter packing, or hydrotropic effects that increase surfactant partitioning to the interface, even at lower overall surfactant concentrations compared to that of bulk CMC.

# 3.3 Hygroscopicity



Given the atmospheric relevance of surfactants in potentially lowering the energy barrier for water condensation (via surface tension reduction), we conducted measurements to assess how the presence of trace amounts of OSs modifies droplet hygroscopicity under subsaturated conditions. We trapped a single particle at RH  $\approx 10\%$  and monitored its size growth using Raman spectroscopy as the RH was gradually increased at a rate of approx. 1% RH per 90 s, up to RH  $\approx 94\%$ . We note potential lim-

itations in our RH sensor accuracy above ≈ 94%; reported values in this range may be underestimated or slow to equilibrate.

Experiments were repeated at least 5 times for each ternary system (CA:SMS 10:1, CA:SES 10:1, CA:SOS 100:1, CA:SDS 1000:1), using droplets of varying initial sizes (between 5–10 μm diameter).

**Figure 6.** Influence of surfactant composition on the hygroscopic growth of citric acid droplets. (a) Droplet radius versus relative humidity (RH) for pure citric acid (CA) and CA mixed with different surfactants at specified molar ratios (CA:SMS 10:1, CA:SES 10:1, CA:SOS 100:1, CA:SDS 1000:1). (b) Corresponding growth factor (GF) curves, calculated relative to the size at RH = 10%. (c) Hygroscopicity parameter ( $\kappa$ ) derived from the growth curves using Eq. (10), plotted as a function of RH.

Figure 6(a) depicts the radius of droplets containing these ternary systems as a function of the cell's RH. Despite different initial sizes, all systems exhibit very similar growth behavior. We calculate the growth factor (GF) using:

$$GF = \frac{d_{\text{wet}}}{d_{\text{dry}}},\tag{9}$$

where  $d_{\text{wet}}$  is the particle's wet diameter at a given RH, and  $d_{\text{dry}}$  is the particle's diameter measured at the starting RH of 10% (dry conditions). As seen in Figure 6(b), the GF curves for all ternary systems are almost identical to that of the binary CA system. They show smooth growth up to RH  $\approx 80\%$  (GF  $\approx 1.2$ ), followed by more rapid water uptake, reaching GF  $\approx 1.8$  at RH  $\approx 94\%$ . Slight differences at the highest RH are likely within experimental uncertainty, possibly including sensor limitations. GF measurements using different initial droplet sizes yielded consistent results for each system.

Furthermore, we calculate the hygroscopicity parameter ( $\kappa$ ) of these systems, a key parameter for assessing cloud droplet activation potential. The  $\kappa$  value is determined at each RH step using the measured GF, the measured surface tension  $\sigma(RH)$  (from Figs. 4b and 5), and the water activity  $a_w$  (assumed  $a_w \approx RH/100\%$ ), via the  $\kappa$ -Köhler equation solved for  $\kappa$  (Petters and Kreidenweis, 2007; Bramblett and Frossard, 2022):

$$\kappa = (GF^3 - 1) \left[ \frac{1}{a_w \exp\left(-\frac{4\sigma M_w}{RT\rho_w d_{wet}}\right)} - 1 \right],\tag{10}$$

where  $M_{\rm w}$  is the molar mass of water and  $\rho_{\rm w}$  is the density of pure water.

**Figure 7.** (a) Efflorescence and deliquescence dynamics of an optically trapped droplet containing NaCl + SES (10:1) at 4 stages. (b) A homogeneous aqueous droplet is initially trapped at RH = 50% with MDRs present on the top of the O-H band in the Raman spectrum. (c) By lowering the RH down to 47%, the droplet effloresces and forms a phase-separated particle with a small amount of water left (note the residual O-H band intensity). (d) While the effloresced particle is still trapped, RH is elevated gradually, and the particle shows water uptake, as indicated by the increasing O-H band intensity at three different RH values; however, it still maintains its phase-separated, partially engulfed form. (e) At RH = 72%, the particle instantly takes up a considerable amount of water from ambient air, deliquesces, and becomes homogeneous again, as MDRs reappear.

Figure 6(c) presents hygroscopicity curves for the OS systems. Consistent with the GF curve, hygroscopicity is nearly identical in behavior for all systems. At high RH, results converge to the value of  $\kappa \approx 0.2$ , which has been previously reported for aqueous tricarboxylic acids like CA, where these literature values were either calculated assuming the surface tension of water (Han et al., 2022) or derived directly from growth factor measurements without surface tension corrections (Marsh et al., 2017). Therefore, based on the observed hygroscopic response, these surfactant molar ratios have no effect on water activity. Conversely, surface tension suppression does not appear to increase or decrease the hygroscopicity under subsaturated conditions in this super-micron size regime. This is reasonable, as hygroscopicity primarily reflects the solute effect (Raoult term), which is largely independent of the curvature (Kelvin) effect that includes surface tension.

## 3.4 Formation of a partially engulfed particle

An aqueous NaCl droplet undergoes efflorescence at approximately 47% RH (Li et al., 2014), typically forming a solid, dry crystal. This was also observed here, however, when a small amount of a surfactant such as SES was introduced, the droplet instead adopted a partially engulfed morphology (Kwamena et al., 2010) upon efflorescence around this RH, retaining some of its water content. Furthermore, the particle remained optically trapped in all experiments with SES present, indicating it retained a more spherical overall shape compared to a dry NaCl crystal (pure NaCl particles typically fall from the trap upon crystallization under these conditions due to their non-spherical crystal morphology). As illustrated in Figure 7, a droplet

comprising NaCl and SES (10:1 molar ratio) is initially maintained at RH = 52% and gradually dried until reaching the efflorescence point at RH = 47%. The presence of sharp MDRs confirms that the particle remains a homogeneous liquid phase before efflorescence, as shown in Figure 7(b). Upon reaching the efflorescence point, the droplet loses most of its water content and adopts a phase-separated, possibly partially engulfed, structure. Figure 7(c) shows that even after efflorescence, the O-H stretching band of water exhibits weak intensity, indicating a small amount of residual water associated with the particle. By subsequently increasing the RH towards 72%, we observe a gradual water uptake (Figure 7(d)) while the particle maintains its phase-separated morphology. As it approaches the deliquescence point (around 72% RH), the particle undergoes a sudden phase transition into a homogeneous state. At this stage, sharp MDRs reappear, indicating that the droplet has regained its spherical liquid shape, as depicted in Figure 7(e). The presence of surfactants here appears to have two effects: 1) It may allow the effloresced particle to maintain sufficient sphericity to remain stable in the optical trap. 2) Solid NaCl has a high surface energy, potentially inhibiting water uptake below the deliquescence RH. The presence of the surfactant, by coating the NaCl core or influencing the interface, likely facilitates water uptake by the particle. A thin interfacial layer of strongly bound water between the NaCl core and the surfactant shell could explain this residual water signal and the retention of a nearspherical shape, consistent with hydration structures reported for NaCl-SDS core-shell systems (Harmon et al., 2010). Thus, instead of only a sudden large water uptake at deliquescence, the ternary particle shows gradual water uptake below the final deliquescence point, eventually becoming fully aqueous (Figure 7(e)). We note that the hysteresis between efflorescence (47%) RH) and deliquescence (72% RH) in our NaCl-SES system, together with the gradual growth during hydration, is similar to the behavior reported for NaCl-SMS mixtures (Estillore et al., 2016), suggesting that OSs generally modify the phase transition behavior of salt particles by stabilizing partially hydrated states.

## 4 Conclusion







In this study we showed how OS surfactants influence the surface tension and hygroscopic properties of aerosol particles, particularly under supersaturated and highly concentrated conditions. We demonstrate that OS surfactants, especially those with longer hydrocarbon chains such as SDS, substantially reduce the surface tension of aerosol droplets even at very low surfactant concentrations, potentially affecting the activation of cloud condensation nuclei and cloud formation processes. Despite significant reductions in surface tension, the hygroscopicity of aerosol particles containing citric acid and trace OSs remains similar to that of binary citric acid systems, indicating that surfactants primarily modify surface properties rather than water uptake under subsaturated conditions. Additionally, the presence of OS surfactants alters particle morphology upon efflorescence, promoting partial water retention and facilitating gradual water uptake prior to complete deliquescence. These results highlight the importance of including surfactant effects on surface tension within atmospheric models to more accurately predict aerosol impacts on cloud formation and climate.

Code availability. The MRFIT source code is available at https://web.meteo.mcgill.ca/~tpreston/code.html

Data availability. All surface tension, water activity, and refractive index measurements presented in this study are available in the Supplement.

Author contributions. V.S. and T.C.P. conceived the study and designed the experiments. V.S. and C.L. conducted the experiments and analyzed the measurements. M.N.C assisted with providing the material. V.S., M.N.C, and T.C.P. wrote the manuscript. T.C.P. supervised the study. All authors read and approved the revised manuscript.

Competing interests. The authors declare no competing interests.

Acknowledgements. V.S. and T.C.P. acknowledge financial support from the Simons Foundation (SFI-MPS-SRM-00005211). H. T. L. and M. N. C. acknowledge financial support from the Hong Kong Research Grants Council (ref no. 14303023 and 14300524).

335

- Bain, A.: Recent advances in experimental techniques for investigating aerosol surface tension, Aerosol Sci. Technol., 58, 1213–1236, https://doi.org/10.1080/02786826.2024.2373907, 2024.
- Bain, A., Chan, M. N., and Bzdek, B. R.: Physical properties of short chain aqueous organosulfate aerosol, Environ. Sci.: Atmos., 3, 1365–1373, https://doi.org/10.1039/D3EA00088E, 2023a.
- Bain, A., Ghosh, K., Prisle, N. L., and Bzdek, B. R.: Surface-area-to-volume ratio determines surface tensions in microscopic, surfactant-containing droplets, ACS Cent. Sci., 9, 2076–2083, https://doi.org/10.1021/acscentsci.3c00998, 2023b.
  - Berry, J. D., Neeson, M. J., Dagastine, R. R., Chan, D. Y., and Tabor, R. F.: Measurement of surface and interfacial tension using pendant drop tensiometry, J. Colloid Interface Sci., 454, 226–237, https://doi.org/10.1016/j.jcis.2015.05.012, 2015.
  - Bramblett, R. L. and Frossard, A. A.: Constraining the effect of surfactants on the hygroscopic growth of model sea spray aerosol particles, J. Phys. Chem. A, 126, 8695–8710, https://doi.org/10.1021/acs.jpca.2c04539, 2022.
  - Brüggemann, M., Xu, R., Tilgner, A., Kwong, K. C., Mutzel, A., Poon, H. Y., Otto, T., Schaefer, T., Poulain, L., Chan, M. N., and Herrmann, H.: Organosulfates in ambient aerosol: State of knowledge and future research directions on formation, abundance, fate, and importance, Environ. Sci. Technol., 54, 3767–3782, https://doi.org/10.1021/acs.est.9b06751, 2020.
  - Bzdek, B. R., Power, R. M., Simpson, S. H., Reid, J. P., and Royall, C. P.: Precise, contactless measurements of the surface tension of picolitre aerosol droplets, Chem. Sci., 7, 274–285, https://doi.org/10.1039/C5SC03184B, 2016.
    - Bzdek, B. R., Reid, J. P., Malila, J., and Prisle, N. L.: The surface tension of surfactant-containing, finite volume droplets, Proc. Natl. Acad. Sci. U. S. A., 117, 8335–8343, https://doi.org/10.1073/pnas.1915660117, 2020.
  - Carslaw, K. S., Lee, L. A., Reddington, C. L., Pringle, K. J., Rap, A., Forster, P., Mann, G. W., Spracklen, D. V., Woodhouse, M. T., Regayre, L. A., et al.: Large contribution of natural aerosols to uncertainty in indirect forcing, Nature, 503, 67–71, https://doi.org/10.1038/nature12674, 2013.
  - Collins, T. J.: ImageJ for microscopy, Biotechniques, 43, S25–S30, https://doi.org/10.2144/000112517, 2007.
  - Darer, A. I., Cole-Filipiak, N. C., O'Connor, A. E., and Elrod, M. J.: Formation and stability of atmospherically relevant isoprene-derived organosulfates and organonitrates, Environ. Sci. Technol., 45, 1895–1902, https://doi.org/10.1021/es103797z, 2011.
  - Eberhart, J. G.: The surface tension of binary liquid mixtures, J. Phys. Chem., 70, 1183–1186, https://doi.org/10.1021/j100876a035, 1966.
- Estillore, A. D., Hettiyadura, A. P. S., Qin, Z., Leckrone, E., Wombacher, B., Humphry, T., Stone, E. A., and Grassian, V. H.: Water uptake and hygroscopic growth of organosulfate aerosol, Environ. Sci. Technol., 50, 4259–4268, 2016.
  - Facchini, M. C., Mircea, M., Fuzzi, S., and Charlson, R. J.: Cloud albedo enhancement by surface-active organic solutes in growing droplets, Nature, 401, 257–259, https://doi.org/10.1038/45758, 1999.
- Fan, W., Chen, T., Zhu, Z., Zhang, H., Qiu, Y., and Yin, D.: A review of secondary organic aerosols formation focusing on organosulfates and organic nitrates, J. Hazard. Mater., 430, 128 406, https://doi.org/10.1016/j.jhazmat.2022.128406, 2022.
  - Flockhart, B. D.: The critical micelle concentration of sodium dodecyl sulfate in ethanol-water mixtures, J. Colloid Sci., 12, 557–565, https://doi.org/10.1016/0095-8522(57)90061-2, 1957.
  - Gantt, B. and Meskhidze, N.: The physical and chemical characteristics of marine primary organic aerosol: a review, Atmos. Chem. Phys., 13, 3979–3996, 2013.
- Gen, M., Hibara, A., Phung, P. N., Miyazaki, Y., and Mochida, M.: In situ surface tension measurement of deliquesced aerosol particles, J. Phys. Chem. A, 127, 6100–6108, https://doi.org/10.1021/acs.jpca.3c02681, 2023.

- Han, S., Hong, J., Luo, Q., Xu, H., Tan, H., Wang, Q., Tao, J., Zhou, Y., Peng, L., He, Y., et al.: Hygroscopicity of organic compounds as a function of organic functionality, water solubility, molecular weight, and oxidation level, Atmos. Chem. Phys., 22, 3985–4004, https://doi.org/10.5194/acp-22-3985-2022, 2022.
- Hansen, A. M. K., Hong, J., Raatikainen, T., Kristensen, K., Ylisirniö, A., Virtanen, A., Petäjä, T., Glasius, M., and Prisle, N. L.: Hygroscopic properties and cloud condensation nuclei activation of limonene-derived organosulfates and their mixtures with ammonium sulfate, Atmos. Chem. Phys., 15, 14 071–14 089, https://doi.org/10.5194/acp-15-14071-2015, 2015.

370

380

- Harmon, C. W., Grimm, R. L., McIntire, T. M., Peterson, M. D., Njegic, B., Angel, V. M., Alshawa, A., Underwood, J. S., Tobias, D. J., Gerber, R. B., et al.: Hygroscopic growth and deliquescence of NaCl nanoparticles mixed with surfactant SDS, J. Phys. Chem. B, 114, 2435–2449, https://doi.org/10.1021/jp909661q, 2010.
- Haywood, J. and Boucher, O.: Estimates of the direct and indirect radiative forcing due to tropospheric aerosols: A review, Rev. Geophys., 38, 513–543, https://doi.org/10.1029/1999RG000078, 2000.
- Heald, C. L., Ridley, D. A., Kroll, J. H., Barrett, S. R. H., Cady-Pereira, K. E., Alvarado, M. J., and Holmes, C. D.: Contrasting the direct radiative effect and direct radiative forcing of aerosols, Atmos. Chem. Phys., 14, 5513–5527, https://doi.org/10.5194/acp-14-5513-2014, 2014.
- Hettiyadura, A. P., Jayarathne, T., Baumann, K., Goldstein, A. H., de Gouw, J. A., Koss, A., Keutsch, F. N., Skog, K., and Stone, E. A.: Qualitative and quantitative analysis of atmospheric organosulfates in Centreville, Alabama, Atmos. Chem. Phys., 17, 1343–1359, https://doi.org/10.5194/acp-17-1343-2017, 2017.
- Iinuma, Y., Böge, O., Kahnt, A., and Herrmann, H.: Laboratory chamber studies on the formation of organosulfates from reactive uptake of monoterpene oxides, Phys. Chem. Chem. Phys., 11, 7985–7997, https://doi.org/10.1039/B904025K, 2009.
  - Intergovernmental Panel on Climate Change: Climate Change 2023: Synthesis Report. Contribution of Working Groups I, II and III to the Sixth Assessment Report of the Intergovernmental Panel on Climate Change, Tech. rep., IPCC, Geneva, Switzerland, ISBN 978-92-9169-164-7, https://doi.org/10.59327/IPCC/AR6-9789291691647, 2023.
  - Jacobs, M. I., Johnston, M. N., and Mahmud, S.: Exploring How the Surface-Area-to-Volume Ratio Influences the Partitioning of Surfactants to the Air–Water Interface in Levitated Microdroplets, J. Phys. Chem. A, 128, 9986–9997, https://doi.org/10.1021/acs.jpca.4c06210, 2024.
  - Kelly, J. T., Bhave, P. V., Nolte, C. G., Shankar, U., and Foley, K. M.: Simulating emission and chemical evolution of coarse sea-salt particles in the Community Multiscale Air Quality (CMAQ) model, Geosci. Model Dev., 3, 257–273, https://doi.org/10.5194/gmd-3-257-2010, 2010.
  - Király, Z. and Dekány, I.: A thermometric titration study on the micelle formation of sodium decyl sulfate in water, J. Colloid Interface Sci., 242, 214–219, https://doi.org/10.1006/jcis.2001.7777, 2001.
  - Kleinheins, J., Shardt, N., El Haber, M., Ferronato, C., Nozière, B., Peter, T., and Marcolli, C.: Surface tension models for binary aqueous solutions: a review and intercomparison, Phys. Chem. Phys., 25, 11 055–11 074, https://doi.org/10.1039/D3CP00322A, 2023.
  - Kleinheins, J., Marcolli, C., Dutcher, C. S., and Shardt, N.: A unified surface tension model for multi-component salt, organic, and surfactant solutions, Phys. Chem. Chem. Phys., 26, 17521–17538, https://doi.org/10.1039/D4CP00678J, 2024.
- 390 Kleinheins, J., Shardt, N., Lohmann, U., and Marcolli, C.: The surface tension and cloud condensation nuclei (CCN) activation of sea spray aerosol particles, Atmos. Chem. Phys., 25, 881–903, https://doi.org/10.5194/acp-25-881-2025, 2025.
  - Kwamena, N.-O. A., Buajarern, J., and Reid, J. P.: Equilibrium morphology of mixed organic/inorganic/aqueous aerosol droplets: Investigating the effect of relative humidity and surfactants, J. Phys. Chem. A, 114, 5787–5795, https://doi.org/10.1021/jp1003648, 2010.

- Köhler, H.: The nucleus in and the growth of hygroscopic droplets, Trans. Faraday Soc., 32, 1152–1161, https://doi.org/10.1039/TF9363201152, 1936.
  - Lac, E. and Homsy, G.: Axisymmetric deformation and stability of a viscous drop in a steady electric field, J. Fluid Mech., 590, 239–264, https://doi.org/10.1017/S0022112007007999, 2007.
  - Langmuir, I.: The constitution and fundamental properties of solids and liquids. II. Liquids., J. Am. Chem. Soc., 39, 1848–1906, https://doi.org/10.1021/ja02254a006, 1917.
- Lee, H. D. and Tivanski, A. V.: Atomic force microscopy: an emerging tool in measuring the phase state and surface tension of individual aerosol particles, Annu. Rev. Phys. Chem., 72, 235–252, https://doi.org/10.1146/annurev-physchem-090419-110133, 2021.
  - Lee, L. A., Reddington, C. L., and Carslaw, K. S.: On the relationship between aerosol model uncertainty and radiative forcing uncertainty, Proc. Natl. Acad. Sci. U. S. A., 113, 5820–5827, https://doi.org/10.1073/pnas.1507050113, 2016.
  - Li, X., Gupta, D., Eom, H.-J., Kim, H., and Ro, C.-U.: Deliquescence and efflorescence behavior of individual NaCl and KCl mixture aerosol particles, Atmos. Environ., 82, 36–43, https://doi.org/10.1016/j.atmosenv.2013.10.011, 2014.

410

- Li, Z., Williams, A. L., and Rood, M. J.: Influence of soluble surfactant properties on the activation of aerosol particles containing inorganic solute, J. Atmos. Sci., 55, 1859–1866, 1998.
- Lienhard, D. M., Bones, D. L., Zuend, A., Krieger, U. K., Reid, J. P., and Peter, T.: Measurements of thermodynamic and optical properties of selected aqueous organic and organic—inorganic mixtures of atmospheric relevance, J. Phys. Chem. A, 116, 9954–9968, https://doi.org/10.1021/ip3055872, 2012.
- Malila, J. and Prisle, N. L.: A monolayer partitioning scheme for droplets of surfactant solutions, J. Adv. Model. Earth Syst., 10, 3233–3251, https://doi.org/10.1029/2018MS001456, 2018.
- Marsh, A., Miles, R. E., Rovelli, G., Cowling, A. G., Nandy, L., Dutcher, C. S., and Reid, J. P.: Influence of organic compound functionality on aerosol hygroscopicity: dicarboxylic acids, alkyl-substituents, sugars and amino acids, Atmos. Chem. Phys., 17, 5583–5599, https://doi.org/10.5194/acp-17-5583-2017, 2017.
- Miller, S. J., Makar, P. A., and Lee, C. J.: HETerogeneous vectorized or Parallel (HETPv1. 0): an updated inorganic heterogeneous chemistry solver for the metastable-state NH<sub>4</sub><sup>+</sup>-Na<sup>+</sup>-Ca<sup>2+</sup>-K<sup>+</sup>-Mg<sup>2+</sup>-SO<sub>4</sub><sup>2</sup>-NO<sub>3</sub><sup>-</sup>-Cl<sup>-</sup>-H<sub>2</sub>O system based on ISORROPIA II, Geosci. Model Dev., 17, 2197–2219, https://doi.org/10.5194/gmd-17-2197-2024, 2024.
- Minerath, E. C., Casale, M. T., and Elrod, M. J.: Kinetics feasibility study of alcohol sulfate esterification reactions in tropospheric aerosols, Environ. Sci. Technol., 42, 4410–4415, https://doi.org/10.1021/es8004333, 2008.
  - Moroi, Y., Motomura, K., and Matuura, R.: The critical micelle concentration of sodium dodecyl sulfate-bivalent metal dodecyl sulfate mixtures in aqueous solutions, J. Colloid Interface Sci., 46, 111–117, https://doi.org/10.1016/0021-9797(74)90030-7, 1974.
  - Mysels, K. J.: Surface tension of solutions of pure sodium dodecyl sulfate, Langmuir, 2, 423–428, https://doi.org/10.1021/la00070a008, 1986.
- Nozière, B., Ekström, S., Alsberg, T., and Holmström, S.: Radical-initiated formation of organosulfates and surfactants in atmospheric aerosols, Geophys. Res. Lett., 37, L05 806, https://doi.org/10.1029/2009GL041683, 2010.
  - Nozière, B., Baduel, C., and Jaffrezo, J.-L.: The dynamic surface tension of atmospheric aerosol surfactants reveals new aspects of cloud activation, Nat. Commun., 5, 3335, https://doi.org/10.1038/ncomms4335, 2014.
- Ovadnevaite, J., Zuend, A., Laaksonen, A., Sanchez, K. J., Roberts, G., Ceburnis, D., Decesari, S., Rinaldi, M., Hodas, N., Fac-430 chini, M. C., et al.: Surface tension prevails over solute effect in organic-influenced cloud droplet activation, Nature, 546, 637–641, https://doi.org/10.1038/nature22806, 2017.

- Peng, C., Razafindrambinina, P. N., Malek, K. A., Chen, L., Wang, W., Huang, R.-J., Zhang, Y., Ding, X., Ge, M., and Wang, X.: Interactions of organosulfates with water vapor under sub- and supersaturated conditions, Atmos. Chem. Phys., 21, 7135–7148, https://doi.org/10.5194/acp-21-7135-2021, 2021.
- Petters, M. D. and Kreidenweis, S. M.: A single parameter representation of hygroscopic growth and cloud condensation nucleus activity, Atmos. Chem. Phys., 7, 1961–1971, https://doi.org/10.5194/acp-7-1961-2007, 2007.
  - Power, R. M. and Reid, J. P.: Probing the micro-rheological properties of aerosol particles using optical tweezers, Rep. Prog. Phys., 77, 074 601, https://doi.org/10.1088/0034-4885/77/7/074601, 2014.
- Preston, T. C. and Reid, J. P.: Determining the size and refractive index of microspheres using the mode assignments from Mie resonances,

  J. Opt. Soc. Am. A, 32, 2210–2217, https://doi.org/10.1364/JOSAA.32.002210, 2015.
  - Prisle, N. L., Raatikainen, T., Laaksonen, A., and Bilde, M.: Surfactants in cloud droplet activation: mixed organic-inorganic particles, Atmos. Chem. Phys., 10, 5663–5683, 2010.
  - Prisle, N. L., Asmi, A., Topping, D., Partanen, A.-I., Romakkaniemi, S., Dal Maso, M., Kulmala, M., Laaksonen, A., Lehtinen, K. E. J., McFiggans, G., and Kokkola, H.: Surfactant effects in global simulations of cloud droplet activation, Geophys. Res. Lett., 39, L05 802, https://doi.org/10.1029/2011GL050467, 2012.

- Rap, A., Scott, C. E., Spracklen, D. V., Bellouin, N., Forster, P. M., Carslaw, K. S., Schmidt, A., and Mann, G.: Natural aerosol direct and indirect radiative effects, Geophys. Res. Lett., 40, 3297–3301, https://doi.org/10.1002/grl.50441, 2013.
- Rassing, J., Sams, P., and Wyn-Jones, E.: Temperature dependence of the rate of micellization determined from ultrasonic relaxation data, J. Chem. Soc., Faraday Trans. 2, 69, 180–185, https://doi.org/10.1039/F29736900180, 1973.
- Robb, I. D.: Determination of the number of particles/unit volume of latex during the emulsion polymerization of styrene, J. Polym. Sci., Part A-1: Polym. Chem., 7, 417–427, https://doi.org/10.1002/pol.1969.150070201, 1969.
  - Santos, S. F., Zanette, D., Fischer, H., and Itri, R.: A systematic study of bovine serum albumin (BSA) and sodium dodecyl sulfate (SDS) interactions by surface tension and small angle X-ray scattering, J. Colloid Interface Sci., 262, 400–408, https://doi.org/10.1016/S0021-9797(03)00109-7, 2003.
- 455 Schmedding, R. and Zuend, A.: A thermodynamic framework for bulk–surface partitioning in finite-volume mixed organic–inorganic aerosol particles and cloud droplets, Atmos. Chem. Phys., 23, 7741–7765, https://doi.org/10.5194/acp-23-7741-2023, 2023.
  - Scott, C. E., Rap, A., Spracklen, D. V., Forster, P. M., Carslaw, K. S., Mann, G. W., Pringle, K. J., Kivekäs, N., Kulmala, M., Lihavainen, H., et al.: The direct and indirect radiative effects of biogenic secondary organic aerosol, Atmos. Chem. Phys., 14, 447–470, https://doi.org/10.5194/acp-14-447-2014, 2014.
- 460 Shahabadi, V., Vennes, B., Schmedding, R., Zuend, A., Mauzeroll, J., Schougaard, S. B., and Preston, T. C.: Quantifying surface tension of metastable aerosols via electrodeformation, Nat. Commun., 15, 1–11, https://doi.org/10.1038/s41467-024-54106-3, 2024.
  - Shakya, K. M. and Peltier, R. E.: Investigating missing sources of sulfur at Fairbanks, Alaska, Environ. Sci. Technol., 47, 9332–9338, https://doi.org/10.1021/es402020b, 2013.
- Sorjamaa, R., Svenningsson, B., Raatikainen, T., Henning, S., Bilde, M., and Laaksonen, A.: The role of surfactants in Köhler theory reconsidered, Atmos. Chem. Phys., 4, 2107–2117, 2004.
  - Stocker, T. F., Qin, D., Plattner, G.-K., Tignor, M., Allen, S. K., Boschung, J., Nauels, A., Xia, Y., Bex, V., and Midgley, P. M.: Climate Change 2013: The Physical Science Basis, Cambridge University Press, Cambridge, United Kingdom and New York, NY, USA, https://doi.org/10.1017/CBO9781107415324, 2013.

- Tolocka, M. P. and Turpin, B.: Contribution of organosulfur compounds to organic aerosol mass, Environ. Sci. Technol., 46, 7978–7983, https://doi.org/10.1021/es300651v, 2012.
  - Vogel, A., Alessa, G., Scheele, R., Weber, L., Dubovik, O., North, P., and Fiedler, S.: Uncertainty in aerosol optical depth from modern aerosol-climate models, reanalyses, and satellite products, J. Geophys. Res.: Atmos., 127, e2021JD035483, https://doi.org/10.1029/2021JD035483, 2022.
- von Szyszkowski, B.: Experimentelle Studien über kapillare Eigenschaften der wässerigen Lösungen von Fettsäuren, Z. phys. Chem, 64, 385–414, https://doi.org/10.1515/zpch-1908-6425, 1908.
  - Wang, Y., Zhao, Y., Wang, Y., Yu, J.-Z., Shao, J., Liu, P., Zhu, W., Cheng, Z., Li, Z., Yan, N., et al.: Organosulfates in atmospheric aerosols in Shanghai, China: seasonal and interannual variability, origin, and formation mechanisms, Atmos. Chem. Phys., 21, 2959–2980, https://doi.org/10.5194/acp-21-2959-2021, 2021.
- Xiong, C., Kuang, B., Zhang, F., Pei, X., Xu, Z., and Wang, Z.: Direct observation for relative-humidity-dependent mixing states of submicron particles containing organic surfactants and inorganic salts, Atmos. Chem. Phys., 23, 8979–8991, https://doi.org/10.5194/acp-23-8979-2023, 2023.
  - Yatcilla, M. T., Herrington, K. L., Brasher, L. L., Kaler, E. W., Chiruvolu, S., and Zasadzinski, J. A.: Phase behavior of aqueous mixtures of cetyltrimethylammonium bromide (CTAB) and sodium octyl sulfate (SOS), J. Phys. Chem., 100, 5874–5879, https://doi.org/10.1021/jp952425r, 1996.
- 485 Yorulmaz, S. C., Mestre, M., Muradoglu, M., Alaca, B. E., and Kiraz, A.: Controlled observation of nondegenerate cavity modes in a microdroplet on a superhydrophobic surface, Opt. Commun., 282, 3024–3027, https://doi.org/10.1016/j.optcom.2009.04.016, 2009.
  - Yu, H., Kaufman, Y. J., Chin, M., Feingold, G., Remer, L. A., Anderson, T. L., Balkanski, Y., Bellouin, N., Boucher, O., Christopher, S., et al.: A review of measurement-based assessments of the aerosol direct radiative effect and forcing, Atmos. Chem. Phys., 6, 613–666, https://doi.org/10.5194/acp-6-613-2006, 2006.