# Peer review of "Surface tension and hygroscopicity analysis of aerosols containing organosulfate surfactants"

_EGUsphere, 2025_

## Author Comment (AC1)

**Reply to RC1:**

1. Page 4, line 111: The authors should clarify why their electrodeformation approach is only applicable at lower relative humidities and is apparently not applicable at higher relative humidities (and therefore pendant droplet measurements are required). The authors should also provide some commentary regarding the different volume scales associated with these two measurement approaches. Given the pendant drop measurements are effectively macroscopic (1-2 mm droplet) whereas the electrodeformation measurements are microscopic (5-10 µm droplet), do the authors need to concern themselves with potential bulk depletion effects in the microscopic measurements that, if not accounted for, could lead to discontinuities between the two approaches? Looking at the concentration ranges explored and comparing the two data sets for sodium methyl and sodium ethyl sulfates, depletion seems unlikely to be important for these systems. However, for the longer chain organosulfates, bulk depletion could be very significant.

The electrodeformation technique itself is not fundamentally restricted to a particular relative humidity (RH) range. However, in our current setup, practical limitations arise from the method used to control RH within the trapping cell, which relies on two mass flow controllers. To minimize perturbations to the optically trapped droplet, the total flow rate must remain low. At the same time, the trapping cell has a relatively large volume, making it difficult and time-consuming to achieve and maintain high RH values (typically above 90%). In addition, the RH sensor employed in the current design tends to underestimate the actual RH, further complicating measurements at the upper RH range. We are currently developing an improved humidity control and sensing system to overcome these limitations and extend the operational RH range of the electrodeformation measurements.

We have added these sentences to explain the limitation:

**During measurements, the cell RH can be stably maintained below 90%. Above this level, achieving stable humidity requires airflow rates that dislodge the droplet from the optical trap, making such conditions impractical with this cell design.**

Regarding the volume differences between the two measurement approaches, we now provide size information for the pendant drop in the manuscript:

**The typical size range of the pendent drop was approximately 2–3 mm at the lower portion of the drop.**

In contrast, droplets probed using the electrodeformation method are in the range of 5–10 $\mu$m (this was already listed in the manuscript).

Regarding bulk depletion effects due to the volume scale difference ($\sim 10^9$-fold between methods), these would be most relevant for long-chain surfactants. For the short-chain OS measurements (SMS, SES), the high concentrations and good agreement between pendant drop and electrodeformation data (Figure 3) suggests depletion effects are minimal at these concentrations. For long-chain OSs, we only studied them in mixtures with CA at very low molar ratios (100:1 to 1000:1), where bulk depletion could potentially be significant. However, the agreement between pendant drop and electrodeformation measurements in Figure 5 suggests that bulk depletion effects are not substantially affecting our measurements, possibly because the presence of CA modifies the partitioning behavior or because both techniques achieve similar surface compositions under these conditions.

2. Figure 2a: sodium methyl sulfate is misspelled in the figure legend.

**Fixed.**

3. Figure 3 and related discussion: This reviewer infers from the experimental section that surface tension and refractive index are simultaneously retrieved in the electrodeformation measurement. The authors should therefore explain why, in Fig. 3b, refractive index measurements extend to lower RH ( 20%) than the surface tension measurements (≈30%).

Surface tension and the refractive index (RI) are retrieved by analyzing measured morphology-dependent resonances (MDRs). Determining surface tension requires electrodeforming the droplet, whereas the RI can be obtained without deformation. Sodium ethyl sulfate becomes highly viscous below 30% RH and the droplet was unstable during electrodeformation but stable during normal optical trapping. Consequently, we could measure the RI below 30% but not the surface tension.

4. Page 9, line 204: It is not clear to this reviewer why the stepwise increase in voltage limits measurements to only highly viscous droplets. What prevents one from setting the relative humidity to 80%, for instance? The authors should more clearly define the underlying factors that limit the range of their electrodeformation approach.

To clarify, that discussion refers to measuring viscosity rather than surface tension using electrode-formation. The limitation is that the CCD detector acquires Raman spectra every 1 s, so relaxation times shorter than 1 s cannot be resolved and viscosity cannot be determined for low-viscosity droplets (e.g., an aqueous organic at 80% RH). The relaxation time expression in the referenced paragraph indicates that viscosities on the order of $10^6$ Pa s or higher yield characteristic times exceeding 1 s. Therefore, these high viscosities should be accessible with this instrument.

5. Figure 4 and related discussion: For sodium ethyl sulfate, the authors note in the discussion of Fig. 3b that surface tension for this system continues to decrease as RH decreases. This also seems to be the case when sodium ethyl sulfate is mixed with NaCl (Fig. 4a). However, when mixed with citric acid (Fig. 4b) the surface tension plateaus at low RH. Do they authors have any explanation for why sodium ethyl sulfate on its own or with NaCl continues to reduce surface tension at lower RH values but plateaus at a value around 60 mN/m when mixed with citric acid?

In Fig. 4b (SES + citric acid), the interfacial composition appears to beome fixed once the RH drops below roughly 60–70%. At these very low humidities, the bulk concentrations of both solutes are already extremely high, so additional water loss no longer changes the surface enrichment (and the surface tension stabilises at approximately 60 mN m$^{-1}$).

The continued decrease seen for the binary SES solution (Fig. 3b) and for SES + NaCl (Fig. 4a) is more unusual. In these droplets, the interface keeps changing even at the lowest RH that we are able to reach, indicating that SES continues to accumulate at the surface. This ongoing enrichment drives the continued fall in surface tension in both cases.

6. Page 10, line 225 (mixing ratios for different surfactants): The authors should provide some additional context describing the motivation for their choices of the various solute:surfactant mole ratios explored in this manuscript. The ratios span 10:1 to 1000:1. Presumably this is due to the relative differences in surface activity for the organosulfates and a desire to study a ratio where clear changes occur, but do the chosen ratios relate in any way to expected compositions in atmospheric aerosol?

Yes, the ratios (10:1 for short-chain, 100:1 to 1000:1 for long-chain OSs) were selected based on the relative surface activities of these compounds, with long-chain surfactants like SDS being significantly more surface-active and thus requiring lower concentrations to produce measurable effects, as well as the need to maintain experimental feasibility (avoiding precipitation and ensuring stable optical trapping).

Regarding atmospheric relevance, while organosulfates can constitute up to 30% of organic aerosol mass in some environments (as noted in third paragraph of the introduction), they typically represent a smaller fraction when mixed with other components like sulfate, nitrate, and non-surfactant organics. Our chosen ratios (0.1-10% surfactant by mole) likely bracket the range found in atmospheric aerosols, though direct composition measurements of mixed OS-inorganic particles remain limited. The key finding that even these trace amounts produce substantial surface tension reduction (30-40%) is atmospherically significant regardless of the exact ratios encountered in nature.

7. Page 12, line 258: The authors can go a bit farther and say explicitly that given the hygroscopic response observed, the surfactant content is insufficient to alter the droplet's water activity.

We have edit the sentence to be more definitive:

**Therefore, based on the observed hygroscopic response, these surfactant molar ratios have no effect on water activity.**

---

## Author Comment (AC2)

**Reply to RC2:**

The manuscript entitled "Surface tension and hygroscopicity analysis of aerosols containing organosulfate surfactants" by Shababadi et al. addresses properties of organosulfates relevant for aerosol water uptake in the atmosphere with a particular focus on surface tension of metastable supersaturated aqueous solution droplets.

Organo sulfates are important constituents of atmospheric aerosols and can influence their properties although little is known in this direction.

The manuscript targets five model sodium salts of organosulfates (sodium methyl sulfate (SMS), sodium ethyl sulfate (SES), sodium octyl sulfate (SOS), sodium decyl sulfate (SDeS), and sodium dodecyl sulfate (SDS) and provides density, refractive index and water activity for aqueous solutions of these organo-sulfates using known methods.

In relation to surface tension, in addition to the pendant drop method, the authors apply a new and novel method (electrodeformation of trapped droplets) to determine surface tension of metastable supersaturated droplets which allow them to determine surface tension in a concentration regime not approached before. This method was applied for the two short chain organosulfates in binary and ternary systems.

The authors also measured hygroscopic growth factors for pure citric acid droplets and citric acid mixed with organosulfates. Finally, experiments were conducted to probe both efflorescence and deliquescence of optically trapped droplets containing SES and NaCl.

The manuscript is timely and presents interesting new results from laboratory experiments using state-of-the-art methods. I have some comments and suggestions for improvements which I find should be addressed before publication. The main ones are:

1) In several places there is room to better use and cite existing literature. Some examples are given below. Please check throughout.

The changes/additions to references are discussed in subsequent comments.

2) the purity of chemicals: some of the chemicals had up to 8% impurity. What could be potential implications of the impurities? This should be discussed.

The potential impact of impurities on surface active molecules should always be considered. Of the organosulfates studied, the most concerning purities are sodium methyl sulfate (SMS, >92%) and sodium octyl sulfate (SOS, >95%).

Based on product specifications, SMS may contain trace amounts of residual alcohols from synthesis. However, such volatile impurities will evaporate rapidly during droplet generation, especially given the use of a nebulization or spraying step. For SOS, a small amount (up to 1%) of 1-octanol may be present due to partial hydrolysis. In our experiments, the sodium octyl sulfate is mixed with citric acid at a 1:100 molar ratio, meaning the resulting 1-octanol concentration is approximately $10^{-4}$ M. At this level, the influence of 1-octanol on the surface tension is expected to be negligible. This estimation is supported by prior studies; for example, see references: Langmuir 1997, 13, 15, 4064–4068 and J. Colloid Interface Sci. 2017, 488, 1–10.

Additionally, our measured surface tension values for sodium octyl sulfate are in good agreement with those reported in the literature, further suggesting that the effect of potential impurities is minimal. See, for example: J. Phys. Chem. B 1996, 100, 50, 19634–19640.

Finally, although sodium dodecyl sulfate (SDS) has a high purity (≥ 99%), we are aware that commercial SDS can contain dodecanol as an impurity, which is strongly surface-active. However, our surface tension measurements are consistent with values reported in the literature for purified SDS samples in which such impurities have been largely removed. This comparison is supported by the findings of the following study: Langmuir 2022, 38, 22, 6794–6801.

Overall, these observations suggest that any surface active impurities present in our samples are at concentrations low enough to have a negligible impact on the reported surface tension values (or simply evaporate prior to the start of the measurement).

3) the last part of the manuscript on hygroscopic growth and efflorescence and deliquescence dynamics is very interesting and provides a good basis for further studies, but it is not as well described as the first part of the manuscript, see further details/suggestions below. I think the manuscript could benefit from including some more details and show more of the available data in 3.3 and 3.4.

The discussion here has been slightly expanded and includes reference to earlier work on deliquescence in salt + organosulfate systems:

**"A thin interfacial layer of strongly bound water between the NaCl core and the surfactant shell could explain this residual water signal and the retention of a near-spherical shape, consistent with hydration structures reported for NaCl–SDS core–shell systems (Harmon et al. 2010)."**

Major comments

Abstract

Use of the word "supersaturated conditions". I suggest clarifying already in the abstract how the authors use the word supersaturation, since I believe this can cause some confusion. When talking about cloud droplets it refers to the saturation ratio of water vapor, however here it refers to the droplet concentration of solute. Perhaps write "in supersaturated aqueous solution droplets" instead of "supersaturated conditions".

The sentence: "We investigate the surface tension and hygroscopicity of aerosols containing short- and long-chain OSs under supersaturated conditions using an electrodeformation method coupled with Raman spectroscopy." was changed. It now reads: **"We investigate the surface tension and hygroscopicity of aerosols containing short- and long-chain OSs in supersaturated aqueous droplets using an electrodeformation method coupled with Raman spectroscopy."**

The last part of the abstract is difficult to follow, for example "the coating takes up water" – but a coating has not been mentioned in the abstract before, and it is unclear at this point what the coating is.

**The word 'coating' has been replaced with 'particle'.**

Introduction

In some cases, I think more original references could be used/included. E.g. Line 31 in relation to partitioning of surfactants between bulk and surface e.g. [1, 2].

**References have been added.**

Line 44: The two references to Wang et al. seem to be in the wrong place. They do not address SDS as a proxy for marine aerosols? one is on terpene derived nitroxy organosulfates (2021a) the

other (2019) on mono-terpene and sesquiterpene derived organosulfates. There are several studies that have used SDS as proxy for marine aerosol e.g. [3, 4].

Thanks for pointing out this error.

**The references have been updated.**

Methods:

The article by Bain et al. 2023[5] provides the same parameters (surface tension, density, refractive index and water activity) for sodium methyl sulfate and sodium ethyl sulfate as in this work. This could be made clearer. Why are slightly different equations used for parameterizations of e.g. density? It would be easier for the reader to compare if the same parameterizations were used.

The density parameterization from Bain et al. 2023 (their Eq. 2) is the same as ours (a quadratic in solute mass fraction). The only difference in presentation is that we immediately set $\rho_0$ to the density of pure water. They parameterized refractive index in terms of solute mass fraction (their Eq. 3), whereas we use molar concentration following Lienhard et al. 2012. We don't know why they used solute mass fraction as they also reference Lienhard et al. 2012 for their parameterizations. Bain et al. did not publish a fitted water activity parameterization (they reported measured water activity and compared it with AIOMFAC).

It would be helpful if it was explained how droplets were generated for the electrodedeformation approach and in what order RH was varied – was is increased or decreased. Also, it could be explained how it was known that the droplets were supersaturated. Were the droplets injected at high RH and then the RH was decreased targeting the efflorescence branch?

Droplet generation is detailed in Shahabadi et al. (2024). Briefly, aerosol droplets are generated via nebulization of the aqueous solution and introduced into the trapping cell, which creates an initially high RH environment. The RH is then lowered over time to reach the target measurement conditions, following the efflorescence branch.

Two criteria establish that the droplets were supersaturated. (i) We compare the measured solute concentrations in our droplets, determined from the known RH–$a_{\text{w}}$ relationships, with established bulk solubility limits. When the concentration exceeds these limits the droplets are definitively supersaturated, as seen with SES droplets reaching > 7 M at 30% RH, well above saturation. (ii) Phase-state confirmation comes from the droplets remaining in a metastable liquid state rather than crystallizing, which we observe directly through our optical measurements, since crystallization would cause dramatic changes in the morphology-dependent resonances and Raman spectra that we monitor. The refractive index also increases sharply at low RH, reaching $n > 1.55$ for SES, which further confirms the highly concentrated nature of these supersaturated droplets.

Regarding surface tension measurements with the Pendent drop tensiometer How was a "stable" drop defined?

A "stable" drop was defined as one that maintained a constant volume without visible oscillations or further elongation, typically after 30-60 seconds of the formation of the droplet, allowing sufficient time for any significant bulk depletion of surfactants to occur.

To further validate stability, we performed additional measurements under two conditions: (i) at ambient room conditions, and (ii) with the droplet sealed in a cuvette filled with the same solution to maintain vapor pressure equilibrium. In both cases, we observed no significant changes or deviations in the measured surface tension values.

Results

 "density measurements reveal that short-chain OS exhibit higher densities at increased concentrations." - this was also reported in Bain et al. and Koda and Namura 1985[6], I suggest rephrasing to acknowledge this.

Here we are presenting measurements that include both short-chain (SMS, SES) and long-chain (SOS, SDeS, SDS) OSs. That sentence is comparing the short-chain results to the long-chain results. Both Koda and Nomura (1985) and Bain et al. (2023) only studied short-chain OSs (specifically SMS, and SES in Bain et al.), so neither reported comparisons between short- and long-chain OS density behavior.

Line 182: why does it say, "in contrast" ? the longer chain ones also show such behavior.

**Removed "In contrast,"**

In the caption to figure 2 it should say what the lines are – I assume they are the fitted lines? I suggest showing previous data in figure 2a for comparison. In Bain et al. the concentration unit was solute mass fraction. The authors could write or include in supporting material what the solute mass fraction was in these experiments to aid comparison.

The caption for Figure 2 now states that "**Solid lines represent fits to the experimental data using the parameterizations given in Equations 1-3 with the best-fit parameters listed in Table 1.**".

The solute mass fraction can be calculated using the density and concentration of the solution, along with the molar weights of the solute and the solvent.

Line 193: it is not clear what reference 2 is.

That error was due to the chosen cite command in LaTeX. It has been fixed.

Table 1 and Figure 2: what are the experimental uncertainties. Can uncertainties on the fitting parameters be provided?

**The accuracies of the instruments used to make the measurements shown in Figure 2 and whose fits are listed in Table 1 are now provided in the main text.** The error bars are much smaller than the points in Figure 2, so they cannot be shown.

Regarding the results on the surface tension of mixtures – it could be interesting to apply some kind of mixing rule. It seems a bit surprising that the surface tension of SES does not stabilize as RH goes down (figure 3) but the surface tension of the mixtures with CA and NaCl does. Why could this be?

Our explanation would be that, for pure SES, as water content decreases with RH, the bulk concentration increases continuously, driving additional surfactant molecules to the interface and reducing the surface tension even at very high concentrations (> 7 M). The absence of a plateau suggests that the surface does not reach saturation even at our lowest measured RH ($\sim$ 33%) and SES can continue to pack more densely at the interface in these supersaturated states.

In contrast, for mixtures with CA or NaCl, the major solute controls the bulk solution properties, while the trace surfactant (10:1 ratio) may more readily saturate the available surface at moderate concentrations. Once surface saturation is achieved, further concentration increases as RH decreases do not substantially change the surface tension, producing the observed plateau. Additionally, interactions between the surfactant and the major solute (CA or NaCl) could influence surfactant partitioning and surface packing, promoting surface saturation at lower bulk surfactant concentrations than in the pure SES system.

Regarding the upper y-axis in figures 4 and 5. At a certain RH - is the concentration of CA the same whether an organosulfate is present or does it apply only to the pure CA solution? If so, this should be stated.

It was previously stated but it likely wasn't as clear as it could have been. The captions to Figure 4 and 5 now respectively include the following sentences: **"The top x-axis indicates the approximate concentration (M) of the major solute (NaCl or CA) in a hypothetical binary aqueous solution without OS at the corresponding RH."** & **"The top x-axis indicates the approximate concentration (M) of CA in a hypothetical binary aqueous solution without OS at the corresponding RH."**

Discussion of Figure 5: explicit values should be given for the "typical surface tensions near the bulk CMC". What are the indications that there could be synergistic interactions – what type of interactions?

The typical surface tensions at bulk CMC are approximately 40 mN/m for SOS and 35-37 mN/m for SDS based on literature values (Yatcilla et al., 1996; Mysels, 1986; Santos et al., 2003), which closely match our measured values at low RH for the CA-OS mixtures, as stated in lines 230-232.

Regarding synergistic interactions, several mechanisms could explain the enhanced surface activity we observe: (i) CA, being a hydroxy-tricarboxylic acid, could interact with the sulfate headgroups of the surfactants through hydrogen bonding, potentially altering their packing at the interface; (ii) the carboxylic acid groups of CA might partially protonate at the interface, reducing electrostatic repulsion between anionic surfactant headgroups and allowing tighter packing; and (iii) CA could act as a hydrotrope, modifying the local environment to enhance surfactant partitioning to the surface. While our measurements clearly show surface tension reduction comparable to bulk CMC values despite much lower total surfactant concentrations, determining the exact mechanism would require additional spectroscopic or molecular-level studies beyond the scope of this work.

In response to this comment we now include more details on possible synergistic interactions: **"The marked surface tension reduction suggests surface saturation may occur, potentially enhanced by CA–surfactant synergistic interactions, such as hydrogen bonding between CA's hydroxyl and carboxyl groups and the surfactant sulfate headgroups, partial protonation of CA reducing electrostatic repulsion and allowing tighter packing, or hydrotropic effects that increase surfactant partitioning to the interface, even at lower overall surfactant concentrations compared to that of bulk CMC."**

Hygroscopicity, ternary systems:

Line 241: "using droplets of varying initial sizes" – these sizes should be given. Did the droplet size matter for the results?

The typical droplet sizes are listed in Section 2.2.2 (between 5 and 10 $\mu$m in diameter), but we now repeat this in that sentence for clarity. Regarding whether droplet size matters for the results: in the subsequent paragraph it is stated, "Despite different initial sizes, all systems exhibit very similar growth behavior."

Figure 6: the actual data points should be given. It should be stated how the lines shown were obtained. What was the reproducibility – it says that the experiments were repeated at least 5 times for each ternary system.

By "5 times," we mean that the hygroscopicity experiments were carried out on 5 different droplets for each ternary system to ensure reproducibility. Each droplet was prepared from a freshly made solution, and each measurement began with a different initial size.

Figure 6(a) shows the result from one representative experiment for each system with a distinct initial droplet size. The other measurements produced similar curves, differing only by a vertical shift due to the starting size. Figure 6(b), on the other hand, is essentially identical across all 5 measurements for each system (25 measurements in total), because the data are normalized to the initial size.

In Figure 6(a), all curves were fitted using spline interpolation and extended to ~24,000 seconds of measurement time. The other panels in Figure 6 are derived from these datasets and illustrate different aspects of the same measurements.

A suggestion: the authors could compare with growth factors predicted using the ZSR mixing rule. Estillore et al. [7] provides growth fators at 85 % RH for SMS and SES – a mixing rule could be used to predict the corresponding growht factor for a mixture of CA ad SMS or SES and compared with the values meaured in this work at the same RH.

Regarding the ZSR mixing rule comparison, while this would indeed be an interesting analysis, we believe our current experimental results already demonstrate the key finding clearly: the hygroscopic growth of our ternary systems (CA with trace amounts of OSs) is essentially identical to that of binary CA systems, as shown in Figure 6. The growth factors and hygroscopicity parameters ($\kappa \sim 0.2$) converge to values consistent with pure CA, indicating that at the molar ratios studied (10:1 for short-chain, 100:1 or 1000:1 for long-chain OSs), the surfactants do not significantly influence water uptake under subsaturated conditions. Given that our measurements show negligible deviation from pure CA behavior, a ZSR prediction would likely confirm this finding without adding substantial new insights to our conclusions.

In the paper by Bains et al. it says "Estillore et al. used a Multi-Analysis Aerosol Reactor System to measure the growth factor for a range of commercially available and synthesised organosulfate aerosol. The authors found that organosulfate aerosol does not undergo efflorescence/deliquescence behavior (except for samples that were suspected to be contaminated with NaCl) and retains an appreciable amount of water even at relative humidities (RHs) below 10%." How does this finding relate to the results in this work?

We observe that binary OS systems, particularly SES, can exist as highly concentrated aqueous droplets even at very low RH (down to ~30%), maintaining a liquid state without crystallization. This aligns with Estillore et al.'s observation that organosulfate aerosols retain appreciable water and do not undergo typical efflorescence/deliquescence transitions. Furthermore, our observation that ternary NaCl-OS systems form partially engulfed structures that retain water after efflorescence (Figure 7) provides additional evidence for the unique water-retention properties of OS-containing particles, though in a different compositional context than the pure OS systems studied by Estillore et al.

**In response to this comment we have added the following sentence: "This behavior is also consistent with previous observations showing that OS aerosols do not exhibit efflorescence/deliquescence transitions (Estillore et al., 2016)."**

3.4: formation of partially engulfed particle.

I find this part of the manuscript highly interesting but also lacking some detail and explanation.

Regarding figure 7: why are there no data between 55 and 70% RH?

There are no sharp MDRs in the water band for that RH range, so we cannot characterize the droplet size using our Mie theory fitting algorithm. However, there are Raman measurements (examples are shown in panel d).

Estillore et al. find a difference between deliquescence and effluresence brances of a NaCl SMS mixture – this could be discussed. Would the authors expect to see something similar for the NaCl SES mixture studied?

Estillore et al. observed hysteresis between deliquescence and efflorescence branches for NaCl-SMS mixtures and a growth factor that smoothly increases with increasing RH prior to deliquescence. Our observations in Figure 7 for the NaCl-SES system are similar: the particle effloresces at 47% RH but doesn't fully deliquesce until 72% RH, with gradual water uptake occurring between these points. This hysteresis, along with the formation of partially engulfed structures that retain water after efflorescence, is consistent with Estillore et al.'s findings and suggests this is a general feature of NaCl-organosulfate mixtures where the surfactant modifies the crystallization and dissolution behavior of the salt.

**In response to this comment we have added the following sentence: "We note that the hysteresis between efflorescence (47% RH) and deliquescence (72% RH) in our NaCl-SES system, together with the gradual growth during hydration, is similar to the behavior reported for NaCl-SMS mixtures (Estillore et al., 2016), suggesting that OSs generally modify the phase transition behavior of salt particles by stabilizing partially hydrated states."**

What is meant with the statement: "Furthermore, the particle often remained opticalled trapped? - this implies that they were not always remaining trapped, please provide some further explanation?

We have revised that sentence to explicitly state that the pure NaCl crystals fall from the trap because of their non-spherical shape, which is in contrast to the SES-containing system.

**"Furthermore, the particle remained optically trapped in all experiments with SES present, indicating it retained a more spherical overall shape compared to a dry NaCl crystal (pure NaCl particles typically fall from the trap upon crystallization under these conditions due to their non-spherical crystal morphology)."**

What exactly is meant with "partially engulfed" ? What is being engulfed and by what and how is that seen from the data? This could be better explained.

The term "partially engulfed morphology" refers to a droplet configuration where the organic surfactant phase incompletely surrounds the inorganic salt crystal while the aqueous phase remains in contact with both phases, as systematically described by Kwamena et al. (2010). **We have added this reference directly after the term in the revised manuscript to guide readers to the detailed morphological descriptions.**

As I understand the authors tried to study surface tension of binary droplets containing salt and longer chain organo-sulfates but this was not technically possible. The reason is not entirely clear from the text. What is meant with "instability in aqueous systems" (line 223)?

The "instability in aqueous systems" refers to precipitation and phase separation that occurred when we attempted to prepare solutions of NaCl with long-chain organosulfates.

We have edited that sentence to read: **"Additionally, NaCl induced instability (i.e., precipitation, phase separation) in aqueous systems with long-chain OSs below RH ≈ 80%."**

1. Li, Z., A.L. Williams, and M.J. Rood, Influence of Soluble Surfactant Properties on the Activation of Aerosol Particles Containing Inorganic Solute. Journal of the Atmospheric Sciences, 1998. 55(10): p. 1859-1866.

2. Sorjamaa, R., et al., The role of surfactants in Köhler theory reconsidered. Atmos. Chem. Phys., 2004. 4(8): p. 2107-2117.

3.  Prisle, N.L., et al., Surfactants in cloud droplet activation: mixed organic-inorganic particles. Atmospheric Chemistry and Physics, 2010. 10(12): p. 5663-5683.

4. Gantt, B. and N. Meskhidze, The physical and chemical characteristics of marine primary organic aerosol: a review. Atmos. Chem. Phys., 2013. 13(8): p. 3979-3996.

5. Bain, A., M.N. Chan, and B.R. Bzdek, Physical properties of short chain aqueous organosulfate aerosol. Environmental Science: Atmospheres, 2023. 3(9): p. 1365-1373.

6. Koda, S. and H. Nomura, Aqueous solutions of sodium methylsulfate by Raman scattering, NMR, ultrasound, and density measurements. Journal of Solution Chemistry, 1985. 14(5): p. 355-366.

7. Estillore, A.D., et al., Water Uptake and Hygroscopic Growth of Organosulfate Aerosol. Environmental Science & Technology, 2016. 50(8): p. 4259-426

---

## Author Comment (AC3)

**Reply to CC:**

Referee report for "Surface tension and hygroscopicity analysis of aerosols containing organosulfate surfactants"

The manuscript provides surface tension measurements of a series of sodium alkyl sulfates (SMS, SES, SOS, SDeS and SDS) in binary and ternary (mixtures with citric acid or NaCl) performed using electrodeformation of optically trapped droplets probed with Raman spectroscopy. The measured values are used to infer the hygroscopic growth factors and kappa-parameters. As the members of the studied series are often used proxies for real atmospheric surfactants – also others than atmospheric organosulfates, as SDS has been used in this context at least since Li et al. (1998) – and also the citric acid can be seen as a proxy various highly-oxidised organic molecules, the reported measurements are relevant for the readers of this journal and especially the study of a series of homologous molecules provides molecular-level insight. The manuscript is well written and suitable for publication in ACP after following comments have been considered.

The major concern comes from the stated purities of the compounds used, especially SMS (>92%) and SOS (>95%): as even a small amount of, say, SDS could significantly affect the surface tensions here, can it be confirmed that no strongly surface-active impurities are present?

The potential impact of impurities on surface active molecules should always be considered. As is pointed out, the stated purities of sodium methyl sulfate (SMS, >92%) and sodium octyl sulfate (SOS, >95%).

Based on product specifications, SMS may contain trace amounts of residual alcohols from synthesis. However, such volatile impurities will evaporate rapidly during droplet generation, especially given the use of a nebulization or spraying step. For SOS, a small amount (up to 1%) of 1-octanol may be present due to partial hydrolysis. In our experiments, the sodium octyl sulfate is mixed with citric acid at a 1:100 molar ratio, meaning the resulting 1-octanol concentration is approximately $10^{-4}$ M. At this level, the influence of 1-octanol on the surface tension is expected to be negligible. This estimation is supported by prior studies; for example, see references: Langmuir 1997, 13, 15, 4064–4068 and J. Colloid Interface Sci. 2017, 488, 1–10.

Additionally, our measured surface tension values for sodium octyl sulfate are in good agreement with those reported in the literature, further suggesting that the effect of potential impurities is minimal. See, for example: J. Phys. Chem. B 1996, 100, 50, 19634–19640.

With respect to sodium dodecyl sulfate (SDS), we are aware that commercial SDS can contain dodecanol as an impurity, which is strongly surface-active. However, our surface tension measurements are consistent with values reported in the literature for purified SDS samples in which such impurities have been largely removed. This comparison is supported by the findings of the following study: Langmuir 2022, 38, 22, 6794–6801.

Overall, these observations suggest that any surface active impurities present in our samples are at concentrations low enough to have a negligible impact on the reported surface tension values (or simply evaporate prior to the start of the measurement).

Minor issues:

1. Line 5 (abstract): "the surface tension continues decrease" – I assume that this refers to the hydrocarbon length, but I can be wrong as well. Please reformulate this sentence to be less ambiguous.

The sentence refers to the behavior of short-chain organosulfates, whose surface tension continues

to decrease with decreasing relative humidity, even under highly viscous and dry conditions. This suggests that surface adsorption persists without reaching a plateau, in contrast to the behavior typically observed for organic acids.

However, the original sentence was awkwardly written. We have revised the sentence in the abstract as follows:

**"For droplets containing short-chain OSs, the surface tension decreases as relative humidity (RH) decreases, even under dry and highly viscous conditions."**

2. Line 20: Perhaps the most recent IPCC assessment report could be cited instead?

We have added a citation to the IPCC's sixth assessment report (2023).

3. Line 39: Study of Li and Jang (2013) did not pass the peer review for ACP (as apparent from the reference), please reconsider this.

We removed this reference.

4. Line 56: For Kelly et al. (2009), please cite the corresponding GMD article instead of the preprint, if there is no special reason for the latter.

Fixed.

5.Line 96: For consistency, please indicate also the instrument(s) used for gravimetry.

We just weigh 5 mL of each sample using a micropipette (Sartorius AG, Germany) and a balance (Sartorius AG, Germany).

We have edited the referenced sentence to include these details:

**"The density ($\rho$) of each bulk solution was determined gravimetrically by weighing 5 mL aliquots dispensed with a micropipette (Sartorius AG, Germany) on a digital balance (Sartorius AG, Germany) accurate to $10^{-4}$ mg."**

6. Equations (1)-(3) and Table 1: These kinds of data are very valuable to test different hypothesis and compare theories of CCN activation. To facilitate it, authors have kindly included the measurement data as a supplement. However, also reporting measurement uncertainties (either in the supplement or at least in Fig. 2) would further facilitate such use.

The accuracies of the instruments used to make the measurements whose fits are listed in Table 1 are now provided in the main text.

7. Line 115: Please provide also original references for the Szyszkowski-Langmuir and Eberhardt models.

References have been added.

8. Equation (5): It might be good to stress that the mole fraction here is the total mole fraction of the solution, as a similar type of equation has been used in related context (e.g. Bzdek et al., 2020) with surface mole fractions.

We now explicitly state that $x_i$ is the bulk mole fraction of the solute.

9. Please provide a reference for Eq. (7).

Reference have been added. It is: Lac, E. and Homsy, G. Axisymmetric deformation and stability of a viscous drop in a steady electric field, J. Fluid Mech., 590, 239–264.

10. Lines 256-261: The conclusions hold, as the literature values refer either to kappas calculated assuming the surface tension of water (Han et al., 2022) or deduced directly from measured growth factors (Marsh et al., 2017). This could be spelled out clearly.

The referenced sentence has been rewritten to clarify this point. It now reads: **"At high RH, results converge to the value of $\kappa \approx 0.2$, which has been previously reported for aqueous tricarboxylic acids like CA, where these literature values were either calculated assuming the surface tension of water (Han et al., 2022) or derived directly from growth factor measurements without surface tension corrections (Marsh et al., 2017)."**

11. Figure 7 (c) and (d) and related discussion: Could at least part of the water signal originate from an adsorbed layer instead of residual water within the particle? Such adsorbed layers have been extensively studied for nanosized NaCl and NaCl-SDS particles (e.g. Harmon et al., 2010, and references therein).

In response to this comment, we have added the following to our explanation in the manuscript:

**"A thin interfacial layer of strongly bound water between the NaCl core and the surfactant shell could explain this residual water signal and the retention of a near-spherical shape, consistent with hydration structures reported for NaCl–SDS core–shell systems (Harmon et al. 2010)."**

Very minor issues:

Line 36: From US to China is not more than half of the Northern Hemisphere, thus "globally" sounds like an overstatement. Maybe either part of the sentence could be reworded?

We have modified the sentence as follows:

**"Organosulfates (R-OSO$_3^-$, OSs) are among the primary and most abundant anionic surfactants in atmospheric aerosols and have been detected worldwide, from North and South America to Asia."**

I know that I am already presenting a minority view at this point, but I would prefer authors to make a clear distinction between "aerosol" and "aerosol particle".

Thank you for raising this point. We agree that the distinction is important. Strictly speaking, "aerosol" refers to the suspension of particles in a gas, whereas "aerosol particle" designates the individual particulate matter within that suspension. In the laboratory, we generate surrogates that mimic atmospheric aerosols but are not perfect representations of ambient particles. Throughout this manuscript, the term "aerosol droplets" is used to describe liquid-phase particles produced in the lab (via a medical nebulizer), and we might have used it interchangeably with the term "aerosols" multiple times.

Additional references:

Harmon, C W., R. L. Grimm, T. M. McIntire, M. D. Peterson, B. Njegic, V. M. Angel, A. Alshawa, J. S. Underwood, D. J. Tobias, R. B. Gerber, M. S. Gordon, J. C. Hemminger, and S. A. Nizkorodov, 2010: Hygroscopic Growth and Deliquescence of NaCl Nanoparticles Mixed with Surfactant SDS. J. Phys. Chem. B, 114, 2435-2449.

Li, Z., A. L. Williams, and M. J. Rood, 1998: Influence of Soluble Surfactant Properties on the Activation of Aerosol Particles Containing Inorganic Solute. J. Atmos. Sci., 55, 1859–1866.